# Antimicrobial Properties of Plant Fibers

**DOI:** 10.3390/molecules27227999

**Published:** 2022-11-18

**Authors:** Lizbeth Zamora-Mendoza, Esteban Guamba, Karla Miño, Maria Paula Romero, Anghy Levoyer, José F. Alvarez-Barreto, António Machado, Frank Alexis

**Affiliations:** 1School of Biological Sciences & Engineering, Yachay Tech University, Urcuquí 100119, Ecuador; 2Colegio de Ciencias e Ingenierías, Universidad San Francisco de Quito (USFQ), Departamento de Ingeniería Química, Quito 170901, Ecuador; 3Colegio de Ciencias Biológicas y Ambientales COCIBA, Instituto de Microbiología, Universidad San Francisco de Quito (USFQ), Laboratorio de Bacteriología, Quito 170901, Ecuador

**Keywords:** plant fibers, antimicrobial properties, biomedical applications, cellulose

## Abstract

Healthcare-associated infections (HAI), or nosocomial infections, are a global health and economic problem in developed and developing countries, particularly for immunocompromised patients in their intensive care units (ICUs) and surgical site hospital areas. Recurrent pathogens in HAIs prevail over antibiotic-resistant bacteria, such as methicillin-resistant *Staphylococcus aureus* (MRSA) and *Pseudomonas aeruginosa*. For this reason, natural antibacterial mechanisms are a viable alternative for HAI treatment. Natural fibers can inhibit bacterial growth, which can be considered a great advantage in these applications. Moreover, these fibers have been reported to be biocompatible and biodegradable, essential features for biomedical materials to avoid complications due to infections and significant immune responses. Consequently, tissue engineering, medical textiles, orthopedics, and dental implants, as well as cosmetics, are fields currently expanding the use of plant fibers. In this review, we will discuss the source of natural fibers with antimicrobial properties, antimicrobial mechanisms, and their biomedical applications.

## 1. Introduction

Healthcare-associated infections (HAI), or nosocomial infections, include contaminations acquired by patients in the hospital, but symptoms usually appear after surgical procedures or during the recovery [1]. They represent a serious public health problem, with a high impact on the mortality rate and quality of life, thereby becoming a worldwide concern and priority. HAIs are associated with medical devices and present a significant economic burden on the public health system in developed and developing countries. HAI rates in ICUs in high-income countries are 5–10%, which is 2–10 times higher in low- and middle-income countries [2,3].

In Europe, according to data from The Healthcare-Associated Infections Surveillance Network, in its 2017 Epidemiological Annual Report, from 2014 to 2017, a statistically significant increasing trend of HAIs in Surgical Site Infection (SSI) procedures was observed. In 2017, 8.3% (11,787) of patients who stayed in intensive care units (ICU) for more than two days had at least one HAI [4]. This problem becomes even more relevant in low- and middle-income countries, as they face greater barriers and additional risk factors due to the lack of human resources, lack of medical supplies and disinfection, inefficiency in infection control, little training, and hospital staff continuous training [5]. Among Device-Associated Healthcare-Associated Infections (DA–HAI), it is common to find bloodstream and urinary tract infections associated with catheters, ventilator-associated pneumonia, and surgical site infections (SSI) due to sutures or implants [6]. The incidence of DA–HAI depends on different factors, such as frequency and duration of device use, infection control practices in the hospital, and immune status of patients [7]. Specifically, HAIs are derived from four factors: the patient, a foreign material (e.g., implants), the infectious agent, and the environment. The recurrent pathogens in HAIs are saprophytic and commensal microorganisms with the potential to become opportunistic pathogens commonly found on the skin, the oral and nasopharyngeal cavities, lungs, the vagina, the large intestine, or colon. The pathogens can spread and develop under suitable conditions [7], where the patients eventually enter in contact with contaminated surfaces or objects (fomites). In the 1980s, HAIs were mainly caused by Gram-negative bacteria such as *Escherichia coli* and *Klebsiella pneumoniae* but antibiotic resistance and the increased use of plastic medical devices have increased bacterial infections recently. According to the Centers for Disease Control and Prevention (CDC), carbapenem-resistant *Enterobacteriaceae* (CRE), methicillin-resistant *Staphylococcus aureus* (MRSA), extended-spectrum ß-lactamases-producing *Enterobacterales* (ESBL-E), vancomycin-resistant *Enterococci* (VRE), multidrug-resistant *Pseudomonas aeruginosa* (MDRPA), and multidrug-resistant *Acinetobacter* species (MDRAs) are considered infectious agents [8].

Currently, several mechanisms have been investigated to eradicate the incidence of DA-HAI caused by multidrug-resistant pathogens (MDR). Many of the developed methods are to incorporate antibacterial properties or encapsulate antibiotics in biomedical devices, as well as to prevent the adhesion of bacteria on them. Among the most common mechanisms are polymer coatings, nanoparticle deposition, and encapsulation within the material [9]. The first seeks to make devices with polymer films, which are synthesized as an anti-infective, antimicrobial, and biocompatible coating on a substrate [10]. The second includes nanoparticles in a multilayer coating on the surface of the devices. The last includes the layer-by-layer technique assemblies and modifies the surface to encapsulate drugs, thus giving an antibiotic property to a substrate [9]. These mechanisms provide a possible solution to the proliferation of bacteria in medical devices but could also represent a risk to patients. The principal reason is their synthetic origin components, nanomaterials, and polymers with non-biodegradable characteristics. They can also produce inflammatory responses or cell death. Therefore, natural alternatives such as plant fibers are an option to eliminate possible side effects [11]. In this review, we discuss the antimicrobial properties of natural fibers as alternatives to prevent bacterial adhesion and inhibit their proliferation. In addition, we comment on some biomedical applications of natural fibers in medical textiles, orthopedics, cosmetics, and tissue engineering.

## 2. Pathogens in Biomedical Devices

Microbial infection is a prevalent issue among biomedical devices, both during routine procedures and surgical interventions. This problem is currently increased by the frequent use of catheters, surgical equipment, sutures, or implants needed to treat several medical conditions [12]. In fact, bacteria are the most common type of microorganism causing worldwide morbidity due to acute and chronic infections [13]. Furthermore, there is an alarming growth rate of infections because of MDR bacteria generated by the overuse of antibiotics and other factors that facilitate their development, such as persistent colonization in the facilities and biofilm mode of growth, among others [14]. The bacteria frequently related to biomedical devices are *E*. *coli*, *Klebsiella pneumoniae*, *Staphylococcus aureus*, *Staphylococcus epidermidis*, *Streptococcus viridans*, *Enterococcus faecalis*, *Proteus mirabilis*, and *Pseudomonas aeruginosa* [15].

Nevertheless, there is a difference in the risk of bacterial infections between developed countries and undeveloped ones. The antibiotic resistance process is a major concern in low- and middle-income countries (LMICs). Due to a variety of available antibiotics in drugstores and poor sale regulation systems, the spread of MDR bacteria is a significant problem for these countries [16]. Contrary to high-income countries, where the regulations for antibiotic sale are strict, LMICs are more vulnerable to the appearance of more aggressive and diverse MDR bacteria [17]. The following section will specifically discuss *E. coli*, *P. aeruginosa*, *S. aureus*, and *S. epidermidis*, which are the most common bacteria found in biomedical device infections, as well as briefly review viral and fungi infections.

### 2.1. Escherichia coli

*E. coli* is the most common Gram-negative microorganism isolated from SSIs, being associated with severe morbidity and mortality rates [18,19]. In addition to SSIs, *E. coli* biofilm formation on biomedical devices is responsible for some infections in patients due to their frequent use. These acquired infections usually occur in the bladder and urinary tract [20]. Even though humans have *E. coli* as a commensal bacterium in their gastrointestinal tracts, and they help to regulate metabolism, there are other harmful strains, so-called *E. coli* pathotypes, responsible for numerous and severe infections, more exactly enterotoxigenic *E. coli* (ETEC), enteropathogenic *E. coli* (EPEC), enteroaggregative *E. coli* (EAEC), enterohemorrhagic *E. coli* (EHEC), enteroinvasive *E. coli* (EIEC), and diffusely adherent *E. coli* (DAEC) [21]. Because *E. coli* is a Gram-negative bacterium, it is resistant to several antibiotics, which represent a high infection risk in the case of pathogenic strains and new emerging clones [22]. Moreover, specific *E. coli* strains are associated with infection of materials such as shunts, urethral and intravascular catheters, and prosthetic grafts and joints [23]. To overcome this issue, several studies are providing new methodologies to avoid *E. coli* biofilm formation. Therefore, it is wise to focus on new natural fibers as biomedical materials capable of inhibiting bacterial adherence and proliferation, so as it prevents infections [24].

### 2.2. Pseudomonas aeruginosa

*Pseudomonas aeruginosa* is a Gram-negative bacterium and can cause infections in both immunocompromised and immunocompetent hosts. Due to its multiple antibiotic resistance and extreme versatility against immune responses and clinical treatments, this bacterium is an organism that can hardly be treated in contemporary clinical practice [25].

*P. aeruginosa* infections are commonly found in immunocompromised patients who used invasive devices such as endotracheal tubes or indwelling catheters, because this microorganism can form biofilms in these devices [26]. Therefore, several mechanisms of some *Pseudomonas* species have been studied to characterize their intrinsic resistance to multiple antibiotics, their efflux systems, and their antibiotic-inactivating enzymes [27]. The ability of *Pseudomonas* to develop biofilms is their main mechanism of virulence, which causes ineffective clinical treatment in the hosts and resistance against their immune responses [25,27]. Therefore, this capacity is critical for patients suffering from cystic fibrosis, who acquire this infection mainly in health centers.

It is well known that doctors should avoid prescribing antibiotics unless necessary to prevent the emergence of resistant strains of *Pseudomonas*. Therefore, the prevention of *Pseudomonas* infections, especially in the hospital setting, avoids huge rates of nosocomial infection among patients [25,26]. However, due to the adaptable nature of the strains, the best approach is to prevent the initial adhesion and the colonization of this bacterium in medical devices [28]. To achieve this purpose, the use of natural fibers in biomedical materials is again proposed as a viable tactic for inhibiting bacterial adherence and spreading in health centers.

### 2.3. Staphylococcus aureus

*Staphylococcus aureus* is a Gram-positive and aerobic bacterium that can adapt to various environments, causing a series of infections and diseases [29]. This bacterium is present in approximately 30% of the healthy human population, either in their skin or nasopharyngeal membranes, being part of the normal microbiota. This bacterium does not cause infections as long as the immune system is reinforced [30]. Depending on the *S. aureus* strains involved and the site of infection, certain strains, such as methicillin-resistant *Staphylococcus aureus* (MRSA), are considered primary pathogens, which cause invasive infections or toxin-mediated diseases [31]. Nonetheless, if it crosses into the bloodstream and somehow gets into internal tissues, *S. aureus* can cause significant health problems, from mild skin infections to severe life-threatening systemic diseases [31]. In fact, *S. aureus* is the primary cause of skin and soft-tissue infections (e.g., cellulitis, impetigo, furuncles, folliculitis, and carbuncles) because the primary way of transmission of this microorganism is by direct contact, such as skin-to-skin, or from contact with contaminated objects [32]. Consequently, despite being a widespread bacterium across the population, under certain conditions and the location of the infection, *S. aureus* can produce health issues, ranging from soft to severe clinical conditions, such as meningitis, endocarditis and urinary tract infections, septic arthritis, pulmonary infections, prosthetic device infections, gastroenteritis, and toxic shock syndrome [33]. Additionally, researchers have used natural cellulose fibers of cotton, which, after functionalization and enhanced hygroscopicity, exhibited bacterial contact inhibition and diffusion inhibition when tested against *S. aureus* [34]. Thus, because *S. aureus* is a prevalent bacterium in wound infections, more research is necessary to find natural alternatives for avoiding its proliferation in wounds.

### 2.4. Staphylococcus epidermidis

The human skin is densely colonized by several different bacteria, archaea, viruses, and fungi [35]. However, *Staphylococcus epidermidis* is a common symbiont bacterium found in healthy human skin. Nevertheless, even though most humans carry the *S. epidermidis* bacteria without presenting infection symptoms, it is the principal reason for nosocomial infection related to invasive procedures [36]. Depending on the context, *S. epidermidis* can help or damage the human skin barrier, being frequently associated with the invasion of the skin or other human barriers via catheter/medical/prosthetic devices [37]. Then, this bacterium can produce biofilms that help to protect them from host defense or antimicrobials [38]. Therefore, despite the widespread presence of *S. epidermidis* on human skin and the evidence suggesting a mutual benefit relationship between skin and bacteria, this microorganism is also the principal reason for human skin infections, being one of the most common nosocomial infections with infection rates as high as those of *S. aureus* [38].

Therefore, *S. epidermidis* is known as the principal nosocomial pathogen related to biomaterial-associated biofilm infections [39]. The main problem is that the bloodstream eventually becomes infected after the sudden release of bacteria from biofilms into surrounding tissues. That is why *S. epidermidis* is present in 22% of the patients with bloodstream infections in intensive care units [40]. Additionally, because of its vast presence across the human skin, staphylococcal biofilm formation is related to a delay in the natural process of re-epithelialization and healing of chronic wounds [41]. Thus, biofilm formation is also one of the main factors for the evolution of infection by *Staphylococcus* species and there are new biomaterials with specific characteristics to solve this issue [39]. The prevention of biofilm formation is essential to avoid infections with *S. epidermidis* when using biomedical devices and during the wound healing process.

### 2.5. Viruses

Antiviral compounds added or contained in natural fibers are valuable in the development of hygienic fabrics for infectious diseases. Among the added compounds are metal nanoparticles, carbon nanotubes, metal oxides and heterostructures with a high degree of efficacy against bacteria, mold, and viruses [42]. In addition, antiviral textiles can inhibit the spread of virus infection and effectively reduce the risk of cross-infection and reinfection. Antiviral materials can inactivate viruses or reduce the surface area of pathogen adhesion [43,44].

Several studies have been conducted on the antiviral efficacy of modified fibers, especially those that were coated with nanoparticles, because it has been observed that these inorganic compounds provide stability and robustness for antimicrobial and antiviral textile nano finishes, at high temperature and pressure, due to their physicochemical characteristics and high coverage of the surface area [42]. For example, Afzal et al. [42] treated a fabric with zinc oxide nanoparticles, suggesting that its antiviral activity resulted from the release of Zn^2+^ ions and other reactive oxygen species that damaged host cell proteins, membranes, and nucleic acids by diffusing into them, causing virus inactivation and cell death. This treated fabric was effective against the herpes virus, influenza, dengue, and hepatitis C with long-lasting antiviral activity, even after 30 wash cycles, suggesting that this fabric could be used in medical equipment to prevent viral transfer. Galante et al. [45] applied a reactive silver ink fabric and a low-surface-energy PDMS polymer to provide the fiber with superhydrophobicity and durable antiviral properties against herpes. This link improved antiviral efficacy and durability compared to silver nanoparticles by having better adhesion and coverage of reactive ionic silver in microfibers. Likewise, Iyigundogdu et al. [46] developed functionalized cotton fibers for antiviral properties against adenovirus type 5 and poliovirus type 1 with positive results. Even functionalized natural fibers have been used for the effective elimination of viruses (MS2) in water as an integral solution with environmental benefits.

### 2.6. Fungi

Fungi are more complicated microorganisms than viruses and bacteria due to their cell eucaryotic structure also being able to be found as yeast and mold that often live in soil and generally are not pathogenic in most healthy people. In fact, most fungi are commensals and certain genera are part of the human microbiota, such as *Candida* spp. [47]. Nonetheless, many fungi can cause hospital-related infections with high mortality rates in patients with compromised immune systems [48,49]. Generally, for human infections, the most common fungi and yeasts are *Candida* and *Aspergillus* spp. [50], as these fungi can spread quickly and damage many organs.

Several plant fibers by themselves or associated with nanoparticles already demonstrated antifungal activity and inhibition of the initial adhesion of opportunistic fungi pathogens’ adhesion [51,52]. Alkan et al. [51] reported different degrees of antifungal activity against *Candida albicans* DSMZ 1386 with silk material separately dyed with madder (*Rubia tinctorium* L.) and gallnut (*Quercus infectoria* Olivier). Arenas-Chávez et al. [52] showed a relevant antifungal activity against *C. albicans* and *Aspergillus niger* through functionalized fabrics, more exactly, cotton natural fiber with nanocomposites based on silver nanoparticles and carboxymethyl chitosan (a natural material derived from the shells of sea crustaceans). Moreover, Okla et al. [53] were able to demonstrate the antifungal activities of the various parts of *Avicennia marina* (a mangrove plant) against *Aspergillus fumigatus* and *C. albicans*. However, little is still known about the applicability of different plant fibers or their several parts by themselves or combined with other antifungal agents (such as metal nanoparticles or other types of natural materials) against the diversity of opportunistic fungi pathogens.

### 2.7. Biofilms

Bacterial biofilms are linked to all nosocomial infections that are mostly associated with devices, which represents a challenge for modern practice. Bacterial cells can coexist in two different forms, in a planktonic state as floating free cells and in a sessile state as cells in biofilms attached to a surface [41]. In this second state, cells demonstrate a phenotypic change with the expression of an exopolysaccharide substance (EPS), commonly known as “silt” production. This expression begins immediately after bacterial adhesion and initial colonization of the surface, leading to the production of a protective barrier of bacteria against the human immune system and therapeutic agents such as antibiotics [54]. Therefore, biofilms are well-defined as complex communities of antibiotic-resistant mono- or multispecies bacteria that reside within an exopolysaccharide matrix after irreversible binding and colonization on a biotic or abiotic surface [55]. Therefore, the medical context is the main source of chronic infections and device-related nosocomial infections.

Both Gram-positive and Gram-negative bacteria can form biofilms in medical devices [56]. The most common are *E. coli*, *P. aeruginosa*, *S. aureus,* and *S. epidermidis*, which are most often found in hospitalized patients, as described in the subsections above. According to the National Institute of Health, these bacterial biofilms are responsible for up to 80% of the total number of microbial infections [13], which include cystic fibrosis, meningitis, chronic wounds that do not heal, endocarditis, and catheters, among others. Although medical personnel have made a continuous effort to maintain a sterile environment in health centers, they are still contaminated by these pathogenic bacteria, making it extremely difficult to eradicate them from surfaces due to their high tolerance against antibiotics and detergents [57]. In addition, biofilms are able to resist host immune responses even when treated with larger or combined antibacterial therapies that exhibit certain biofilm cells called persistent cells, which are inactive cells with low metabolism that can be activated after treatment is over [58]. Therefore, an important approach to address this problem is to prevent the development of biofilms through plant-based fibers as new antimicrobial materials, modification of the surface of the device, and even with local administration of drugs.

## 3. Comparison between Natural and Synthetic Fibers

The focus in the development and research of natural fibers with potential biomedical application is based on characteristics such as lower production cost, renewable, cost-effective, lightweight, and biodegradability [59]. The production of natural fibers is environmentally friendly, contributing to the new generation of sustainable materials and waste reduction [60]. Natural fibers are shown to be a viable biomaterial to replace synthetic fibers due to their composition, sustainable potential, and biological function for biomedical applications [61]. The advantages and drawbacks of natural and synthetic fibers are presented in Table 1:

## 4. Antimicrobial Mechanism in the Vegetable Fibers

The current methods to fight bacterial infections in biomedical devices and implants seek to inhibit biofilm formation by reducing bacterial adhesion on their surfaces or killing bacteria [67]. Predominantly, plant fibers are modified to exhibit two essential characteristics. The first characteristic is a bactericidal effect that causes bacterial death by adding a bioactive molecule [68] to cause cytoplasmic membrane disruption, changes in membrane conductivity, protein synthesis inhibition, and nucleic acid inhibition [69]. The second feature is an anti-biofouling effect that prevents bacterial adhesion to the surface of the fiber [70]. However, the antibacterial effect of plant fibers can be found naturally without any modification, as it was observed in brown-colored cotton fibers due to pigments with tannins content [71]. The fiber extraction process could influence the natural antibacterial properties of plant fibers, considering the removal of carbohydrate and inorganic salts that benefit bacterial growth, changes in pH, and the addition of secondary metabolites that enhance antibacterial function [72]. Antimicrobial agents frequently used in plant fibers are classified as organic and inorganic [68]. Organic agents are natural biopolymers and biomolecules such as chitosan, phenols, alginate, and bioinspired formulations (e.g., antimicrobial peptides, anti-quorum-sensing molecules, and bacteriolytic enzymes) [69]. The most used inorganic agents are metallic nanoparticles, for instance, silver and copper nanoparticles, hydroxyapatite, poly ammonium compounds, antibiotics, and synthetic polymers [70,71]. Surface coating and surface modification are the main strategies to provide antibacterial features for plant fibers. Surface treatments can be achieved by physical, mechanical, and chemical methods. For example, in surface coating, a diversity of antimicrobial agents is loaded onto the device surface and then released over time. The most used surface modification techniques include polymerization and derivatization. Antibacterial agents are adsorbed or immobilized on the surface with polymeric molecules, functional groups, hydrophobic molecules, or nanoparticles. They are immobilized by covalent bonding or radical atom transfer. Examples of these are covalent bonding and hydrophobic polycations of quaternary ammonium salts, single-walled carbon nanotubes, and alkylated polyethyleneimine [73].

In addition, plants already have several bioactive mechanisms to fight against bacterial infections and protect themselves. Those mechanisms can directly affect microorganisms through cytoplasmic membrane disruption, changes in membrane conductivity, and clotting cellular content [74]; or they can indirectly stimulate the release of CD4+ and CD8+ lymphocytes by positive regulation of IL-7 for microbe removal [75]. The antibacterial activity of plants is associated with phytochemicals compounds such as sugars, polypeptides, lectins, quinones, simple phenols and phenolic acids, flavones and flavonoids, terpenoids, tannins, coumarins, alkaloids, cannabinoids, and essential oils. Their chemical structure and hydrophobic and hydrosoluble characteristics have antiseptic action in some cases or can lead to enzyme inactivation, proteins, adhesin bindings, and substrate deprivation to cause bacterial death [76].

Phenolic compounds, such as thymol and carvacrol, extracted from thyme (*Thymus vulgaris*) and oregano (*Origanum vulgare*) have shown effects against *Listeria monocytogenes, S. aureus*, and *E. coli*. Their action is focused on the increment of bacterial cytoplasmic membrane permeability, allowing the release of lipopolysaccharides, and losing their functions as an enzyme matrix, energy transducer, and bacteria’s protective armor [74]. Serrulate-type diterpenoids extracted from *Eremophila neglecta*, *E. serrulata*, *E. sturtii*, and *E. dutonii* have antibacterial activities against some Gram-positive strains, especially methicillin *S. aureus*, which leads to biomedical devices infections. Serrulatanes’ compounds are used as potential coats for biomedical device surfaces avoiding biofilm formation. Serrulatanes’ diterpenoids have been tested against *S. epidermidis* and have shown 99% effectiveness in the prevention of bacterial colonization [67].

## 5. Plant Fiber Composition and Antimicrobial Properties

### 5.1. Hemp Fiber

Bioactive compounds of hemp fibers depend on the variety, but most studies have focused on *Cannabis sativa* L. for biomedical applications [68]. Some components, such as terpenes, flavonoids, polyphenols, esters, lactones, flavonoid glycosides, and cannabinoids, have been identified [77,78]. Among these molecules, cannabidiol is of pharmacological interest due to its antioxidant, antibacterial, and antiproliferative activity [68]. Some chemicals and physical properties are affected by variety, plant age, latitude, soil type, or whether it is derived from a monoecious or dioecious crop [79].

Hemp fiber extraction methods include spray retting, water retting, and osmotic degumming, depending on biomedical applications, because the hemp fiber quality depends on the extraction method [80]. The retting method shows a significant advantage in flexibility and strength properties of the fibers and is suggested to produce high-quality hemp fibers [81]. Some characteristics associated with hemp fibers, such as high tensile strength (550–1110 MPa) and Young’s modulus (30–70 GPa), result in good mechanical properties [82,83]. Pectin (1–3%), lignin (5–12%), cellulose (73–77%), and hemicellulose (7–15%) are present in different proportions, depending on hemp fiber variety and method of extraction [80,81,82]. In addition, some studies [84] reported a higher content of lignin in hemp that contains phenols and aromatics compounds. Some hybrid hemp fibers have been reported with jute and flax to increase the biological applications [85]. The complex composition of hemp fibers is responsible for antibacterial activity [86]. Several studies have reported the antibacterial effect of hemp fibers without and with modification against bacteria (See Table 2). Indeed, some hybrid hemp fibers link the antibacterial functional group with the cellulosic backbone to biopolymer functionalization for biomedical applications [78].

### 5.2. Ramie Fiber

Ramie fibers obtained from *Boehmeria nivea* have demonstrated biocompatibility and low production costs. Nowadays, China, Brazil, and the Philippines are the largest producers of ramie fiber [88]. Ramie fibers are a source of bast fiber for the textile, medicine, and cosmetic industry [89]. High concentrations of phenolic and flavonoid compounds have been found in this fiber, and this might contribute to its antibacterial properties, antioxidant and antiproliferative, as reported by Wang et al. [90]. Biomedical applications of ramie fibers have increased in recent years due to their biomaterial characteristics, such as good tensile strength (400–938 MPa), highest Young’s modulus (44–128 GPa), excellent conductivity, and high biocompatibility [83,91]. Its length and strength are superior to cotton or silk. Kandimalla et al. [92] reported 70% wt cellulose in ramie fiber, as well as hemicellulose, and lignin presence.

In particular, Ramesh et al. [93] reported 70–83% of cellulose content and 5–12.7% of lignin in ramie fiber. Additionally, Romanzin et al. [94] reported that the cellulose content of ramie fiber was 68.6–76.2%, with 13.1–16.7% hemicellulose, 0.6–0.7% lignin, and 1.9% pectin. As a result, cellulose, hemicellulose, and lignin content in ramie fiber depend on the extraction method, plant variety, environmental conditions, and others. Some studies have also identified the antibacterial effect against Gram-positive and Gram-negative bacteria described in Table 3. It is recognized that ramie fibers have inherent bacteriostatic ability characterized by excellent antimicrobial erosion, but their surface functionalization can increase the antibacterial properties [94,95].

### 5.3. Sisal Fiber

Sisal fibers, obtained from *Agave sisalana* leaves, are produced in several tropical countries such as Brazil, Tanzania, Mexico, Haiti, and Venezuela [97]. Because of their fibrous properties and resistance, they are generally used in the textile and agricultural industry [98]. Sisal fibers are considered lignocellulosic fibers, with a chemical composition of up to 20% lignin, 44–88% cellulose, and 15% hemicellulose [99]. Lignin provides support and mechanical resistance, binding material, and improves intracellular union [100]. The mechanical properties of these fibers correspond to a tensile strength value of 507–855 MPa and good Young’s modulus of 9.4–28 GPa [83]. In addition to their mechanical resistance and elasticity, sisal leaves are considered a source of bioactive compounds, such as flavonoids, saponins, and terpenes. Furthermore, several in vitro studies support the antimicrobial properties of agave extract with antibacterial effects against Gram-positive and Gram-negative bacteria, such as *Bacillus stearothermophilus, E. coli, Salmonella typhi,* and *S. aureus,* through mechanisms of cellular rupture, causing changes in the structure of the bacterial membrane, ending with cell death [97,101,102]. Table 4 shows a numerous antibacterial studies based on extracts, non-modified and modified of sisal fibers with effective results. 

### 5.4. Cotton Fiber

Cotton is the most popular consumable natural fiber because of its numerous favorable properties, such as natural appearance, heat transfer, wearing comfort, moisture absorbency, and renewable status. This fiber has been produced in different areas of the world, including China, India, the United States, Pakistan, and Brazil, which are among the biggest producers of cotton fibers in the world [106]. Moreover, among the species of the *Gossypium genus, G. barbadense, G. arboretum,* and *G. herbaceum* are very well known for their good quality and low difficulty to grow, and *G. hirsutum* is the most widespread species of cotton worldwide [107]. Cotton fibers exhibit a tensile strength of 287–840 MPa and a Young’s modulus of 9.4–22 GPa [83,108]. These cotton fibers are mostly used for clothing, and because cloths are in contact with all external microorganisms, they can be microorganism carriers. Furthermore, incorporating antimicrobial properties into clothing materials is valuable for avoiding skin allergies, and new antibacterial cotton fibers are being tested [109]. Mixtures of cotton/bamboo fibers and cotton fibers loaded with nanoparticles are some approaches toward antimicrobial textiles. In vitro studies related to functionalized cotton fibers with nanoparticles (NPs) against bacteria showed that this polar fiber and cationic ions from NP interact with the anionic features of the bacteria cell envelope, causing the denaturation of several proteins of bacterial membranes, causing the bacteriolytic effect in Gram-positive and Gram-negative bacteria [110]. Another coating proved for cotton fibers is the functionalization with antimicrobial proteins to enhance the inhibition of microbial growth with effective results against Gram-positive *B. subtilis* and *E. coli*. Cotton fibers confirmed good to excellent antimicrobial activities, depending on the bacteria (See Table 5). Then, cotton fibers also can be combined with an antifouling or antibiofilm approach, increasing advantages for biomedical applications [111].

### 5.5. Linen or Flax Fiber

Linen fibers are obtained from *Linum usitatissimum* L. and their production and commercialization are from China, Italy, Tunisia, and Lithuana. The low-cost production, low density, and biodegradability are advantages of flax fibers [116]. Their applications are related to textiles, composites, and specialty papers. Linen fibers have antifungal properties (See Table 6). Flax fibers have many advantages such as air permeability and comfort in clothing [117]. These kinds of fibers are appropriate for skin biomedical applications due to their antifungal functionality.

A summary of the antibacterial activity of hemp, ramie, sisal, and cotton fiber are presented in the Figure 1.

## 6. Biomedical Applications of Antimicrobial Fibers

Plant fibers with antimicrobial properties have been demonstrated to have physical, mechanical, and chemical characteristics to qualify as potential materials for biomedical applications [119]. Biocompatibility and minimal toxicity with mammalian tissue, tensile features, and compression strength have been studied to be applied in designing new solutions in the health and cosmetic fields [120]. The following part of this review shows some relevant studies in medical textiles and suture materials, orthopedics and dental implants, and cosmetics applications (See Figure 2).

### 6.1. Natural Fibers on Medical Textiles

Textiles have an essential role in the medical industry, especially for use inside medical facilities. Healthcare workers require functional textiles to work with pathogenic bacteria and viruses; these materials must offer specific properties over traditional textiles, such as antimicrobial, liquid repellant, or hypoallergenic properties [122]. For any healthcare textile, a series of standard properties are required, so these can be employed in the form of hospital apparel (i.e., surgical aprons), face masks, and arm or knee braces [123]. Moreover, these textiles can be fabricated using different polymers and fibers; cotton is among the most common materials used in these applications [124]. Additionally, many fibers are developed for medical purposes, which can be both natural and synthetic. Thus, among the most used natural fiber-forming polymers are cellulose, chitosan, and chitin, and proteins such as collagen and gelatin, alginic acid, and hyaluronic acid [14]. The main characteristics of a medical textile material should be flexibility, biocompatibility, strength, extensibility, air permeability, absorbency, and availability in three-dimensional structures [125]. Nevertheless, the combination of these properties depends on the purpose of the medical textile, so it can be used from simple wound bandages to suture materials during surgery [126].

In addition, because of the multiple benefits of cellulose fibers, they are used in the form of nanocellulose for application in biomaterials, which include skin replacements for different injuries such as burns or wounds, as well as drug-releasing systems. Other natural fibers such as coconut (*Cocos nucifera*) and sisal (*Agave sisalana*) have been used to produce suture material and have been applied to animals to heal wounds, where sisal showed the best results for surgical suture, inhibiting biofilm formation while supporting the correct wound healing process [101]. The fibers from the ramie plant (*Boehmeria nivea*) were also used in another study to produce a natural biocompatible suture biomaterial. This fiber was shown to be biocompatible toward human erythrocytes and nontoxic to mammalian cells in addition to its high efficacy for wound closure [92]. Therefore, natural fibers such as those extracted from sisal or ramie, as well as polymers such as collagen or chitin, have the potential to be used in the medical field as biomedical textiles. In the same way, the *Agave sisalana* plant leaf fibers showed an efficient antimicrobial result in the surgical site with good tensile properties and intrinsic radiopacity retention [127].

Thus, a variety of natural fibers in different combinations and shapes can be used for textile materials such as yarns, fibers, or filaments, depending on the specific application, which ranges from hygienic products such as diapers or sanitary napkins to protective surgical gowns, compression fabrics, sutures, prosthetic materials, or extracorporeal devices [128,129]. Furthermore, there is increasing research on antibacterial cellulose fibers with no antibacterial agents added or further modification to the fibers. For example, studies have recently used the fibers from *Hyptis Suaveolens* to evaluate them against Gram-positive and Gram-negative bacteria in vitro, demonstrating the capability of *Hyptis suaveolens* to avoid bacterial growth on their surface [128]. Fibers can be loaded with bioagents by encapsulation, coating, adsorption by several chemical methods, and controlled release from the fibrous textile [130]. Consequently, the application of natural fibers in the medical field exhibits great potential, and the correct matching of their properties and characteristics will be adjusted depending on the medical application.

### 6.2. Natural Fibers on Orthopedics, Dental Implants, and Cosmetics

Orthopedic areas demand materials with high strength, high stiffness, good flexibility, and energy absorption properties. Among the innovations in orthopedics and prosthetics, natural fibers with antibacterial properties have been directed as a complement to hybrid biocomposites by their mechanical and antibacterial properties [131]. Arumugam et al. [132] used sisal (SF), glass (GF), and chitosan fibers (CTF) for application in the orthopedic field. This recent study assembled GF/SF/CTF composite scaffolds. These scaffolds demonstrated excellent mechanical properties, with considerable compressive and flexural strength improvement. They could be a promising biomaterial for fracture fixation of orthopedic bone plates, mainly used to control bone fracture and reduce fracture space, allowing primary bone healing. The work done by Gouda et al. [133] tested the mechanical properties of Hybrid Natural Fiber Polymer Composite Material (HNFPCM) using epoxy resin as a matrix and including natural fibers of sisal, jute, and hemp. In this study, HNFPCM’s results of tensile tests showed a match with the tensile property of the femur bone; therefore, this material is considered a candidate for the manufacture of orthopedic implants, especially for femur prostheses [133]. For orthopedical implants, a strong antibacterial activity against *S. aureus* and *E coli* is needed [134].

Furthermore, natural fibers are mixed with different polymers to build scaffolds, especially for bone tissue engineering. In a recent study from 2020, silanized sisal fibers were used to fabricate a bimodal 3D scaffold with Poly (ε-caprolactone) and nano-hydroxyapatite (n-HAP), resulting in a two-fold increase in compressive strength and modulus [69]. Bamboo fibers were incorporated into nano-hydroxyapatite/poly(lactic-co-glycolic) by freeze-drying, achieving a scaffold with good cytocompatibility and better degradation rates [135].

Another natural fiber with antibacterial properties used in the orthopedic area is ramie fiber. This fiber has a wide application in the manufacture of intelligent medical textiles due to its antibacterial and mechanical properties. In 2016, Tianjin Polytechnic University proposed using ramie fibers for manufacturing medical splints to treat bone injuries. It will improve the performance of the medical splint and reduce pollution and manufacturing costs at the same time [136]. Based on these investigations, natural fibers such as sisal, jute, ramie, and hemp are suitable candidates for application in the orthopedic area; these fibers provide favorable mechanical and antibacterial properties and an ecological value. Although research is still in its initial phases, taking these biocomposites into account contributes to developing applications in implants, plates, or textiles related to the orthopedic area.

Biomedical applications of natural fibers and their composites in the dental field can be possible due to mechanical properties such as strength and lower mass [137]. Fibers are used in orthodontics, dental implantology, and other dentistry fields. Natural fibers would be combined with a resin matrix to obtain similar results that synthetic fibers in teeth replace. Bamboo and sisal fibers have been studied with good results in compression strength [138]. Fiber requires parameters such as a high elastic modulus and workable matrix [139]. The bioactive restorative fiber for dental application would have antimicrobial activity against *Streptococcus mutans*, enhancing dental restorations and avoiding caries spread [140]. However, natural fibers have some challenges based on high water absorption affecting their performance. For this reason, natural fibers would be used to create hybrids with synthetic fibers and other biomaterials for medical applications. In the cosmetics industry, natural fibers of silk have been used for facial mask paper applied for skin care [141]. Keratin fibers obtained from wood are used in pharmaceutical and cosmetics products [142]. The avocado functional kinetic fiber was developed for skin beauty for female consumers [143]. However, natural fibers need functionalization with protein, immunoglobulins, and polymers with varying synthesis techniques for future biomedical applications [144].

## 7. Conclusions

Healthcare-associated infections (HAI), or nosocomial infections, are a worldwide problem aggravated by the lack of control in hospitals and antibiotic-resistant bacterial strains. The most prevalent bacteria, such as *E. coli* and *S. aureus*, are controlled by coatings of nanoparticles and antibacterial substances with side effects in patients.

Recently, sisal, ramie, and hemp fibers have been studied in biomedical applications due to their morphological bioactive molecules such as cannabidiol, high concentration of phenolic and flavonoid compounds, and antibacterial properties. Current research uses vegetable fibers and other structural compounds to form biocomposites, showing ideal mechanical, biodegradability, and antibacterial properties for their potential use in orthopedics, implants, medical textiles, and tissue engineering. The opportunity to use natural resources to reduce infection rates in surgeries, implants, materials, and medical supplies and their application opens an opportunity for short-term needs such as for cutaneous applications. Despite the publications on the antibacterial properties of natural fibers in biomedical applications, few works have reached in vivo experimentation, which is a necessary step for future applications and an adequate evaluation of the safety of natural fibers.

## Figures and Tables

**Figure 1 molecules-27-07999-f001:**
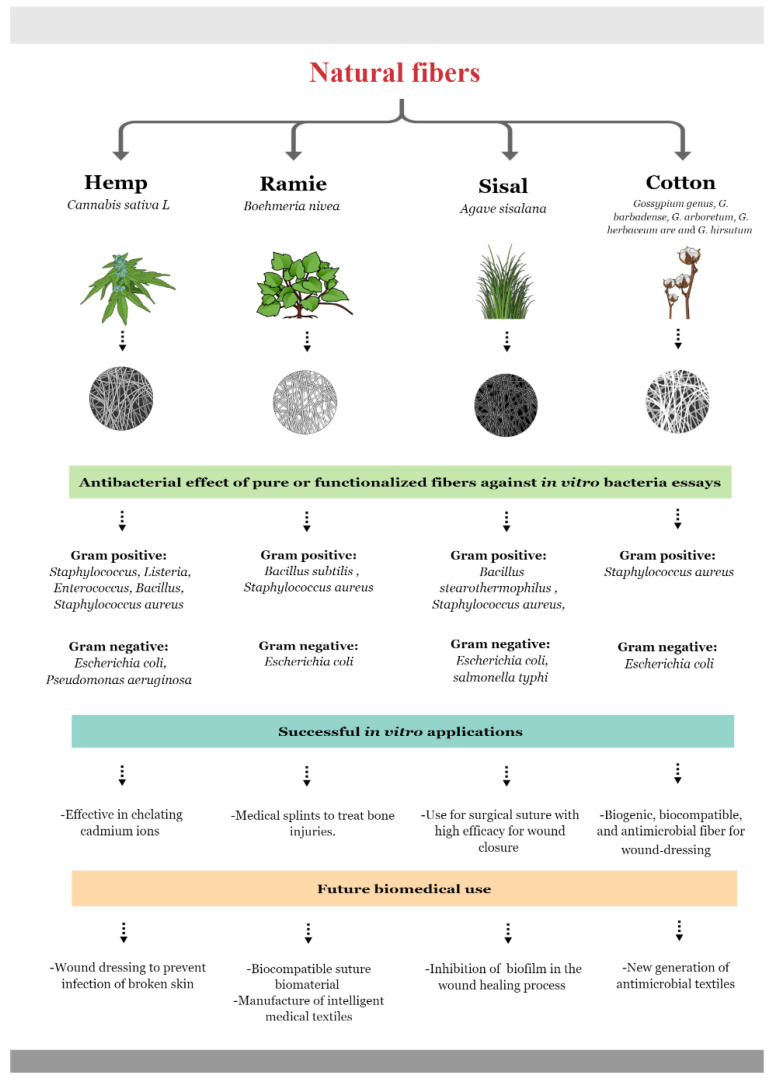
Natural and functionalized fibers’ effect on Gram-positive and Gram-negative bacteria [86,94,102,111] (https://mindthegraph.com/ accessed on 2 October 2022).

**Figure 2 molecules-27-07999-f002:**
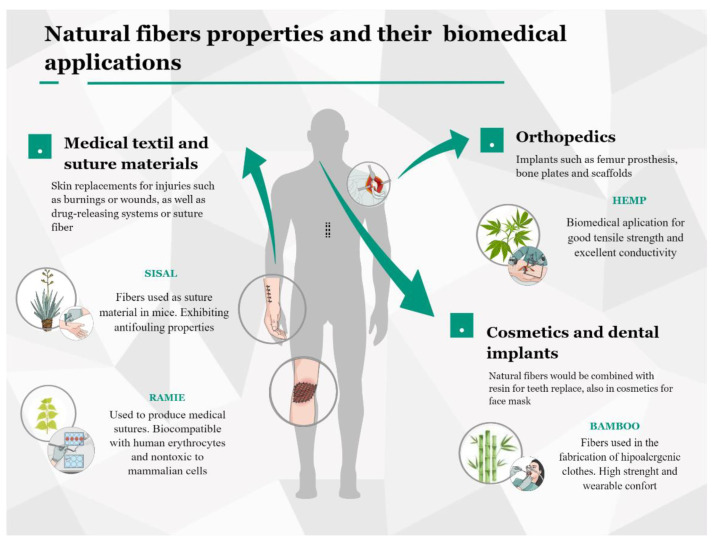
Natural fibers with antimicrobial properties and their biomedical applications [120,121] (https://mindthegraph.com/ accessed on 2 October 2022).

**Table 1 molecules-27-07999-t001:** Natural fibers vs. synthetic fibers.

Characteristic	Natural Fibers	Synthetic Fibers	Refs
Source	It is produced from plants, animals, and minerals	It is manufactured from petroleum-based chemicals.	[60]
Density	It makes the composites lighter because the density is between 1.2 and 1.6 g/cm^3^	It has limited application for composites application by their density (glass fiber = 2.4 g/cm^3^, carbon fiber = 1.9/cm^3^)	[62,63]
Production	Relatively aligned, long and discontinuous fibers	Well-aligned continuous fibers	[62]
Principal compounds	The presence of cellulose, lignin, hemicellulose, and pectin	Formed by joining chemical monomers into polymers	[64]
Mechanical properties	High specific properties related to elastic modulus and strength, but drawbacks such as hydrophilic character and low thermal stability	High thermal stability, high elasticity and durability	[61]
Nature	Hydrophilic	Hydrophobic	[65]
Environmental	It is renewable and recyclable	High durability and cost	[66]

**Table 2 molecules-27-07999-t002:** Antibacterial studies of hemp fibers.

Hemp Modification	Bacteria	Type	Method	Result	Ref
**Hemp Fiber Extracts**					
Essential oils from hemp fiber-type (CBD, α-Pinene, β-Pinene β-Myrcene α-Terpinolene β-Caryophyllene)	*Staphylococcus, Listeria, Enterococcus, Bacillus*	Gram- positive	Agar disk diffusion (ADD) assay: plates with Tryptic Soy Agar, controls were ampicillin and ciprofloxacin Minimum inhibitory concentration (MIC): microwell dilution method was used	ADD: Showed good inhibition of bacterial growth compared with antibiotics. MIC: showed non or lower antibacterial activity on *Staphylococcus*, samples showed anti-*Listeria* activity, good antibacterial activity against *Enterococcus*, and excellent antibacterial activity against *Bacillus*	[78]
**Modified Hemp Fibers**					
Hemp fiber grafted with quaternary ammonium groups	*Escherichia coli*	Gram- negative	Antibacterial activity is based on the change of electrochemical potential by electrostatic reaction reducing the cell membrane releasing cytoplasm substances and cells dissolution	The antibacterial activity was 95.41% and after washing 89.78%	[86]
	*Staphylococcus aureus*	Gram- positive		The antibacterial activity was 99.64% and after washing 91.12%	
Hemp fiber with 2-benzyl-4-chlorophenol	*Staphylococcus aureus*	Gram- positive	Minimum inhibitory concentration (MIC) and Minimum Bactericidal Concentration (MCB) test	This resulted in the death of 99.9 % of bacteria	[87]
	*Pseudomonas aeruginosa*	Gram- negative		This resulted in the death of 99.9% of bacteria	

**Table 3 molecules-27-07999-t003:** Antibacterial studies of ramie fibers.

Ramie Modification	Bacteria	Type	Method	Result	Ref
**Non-modified Ramie Fibers**					
Ramie plant fiber as surgical suture material	*Escherichia coli (MTCC40)*	Gram- negative	Agar plate method: agar plates are prepared with nutrients and a layer of bacteria was added to the plate as well as sterile suture fiber.	Good antibacterial activity with a zone of inhibition of 16 mm	[92]
	*Bacillus subtilis (MTCC441) and Staphylococcus aureus (MTCC3160)*	Gram- positive		*B. subtilis* showed a zone of inhibition of 14 mm and the *S. aureus* strain showed a zone of inhibition of 11 mm	
**Modified Ramie Fibers**					
Ramie Fabric Using Titanium Dioxide Nanoparticles	*Escherichia coli*	Gram- negative	Antibacterial effect by the percentage of bacteria reduction (R%) by the equation: $\%=\frac{W-Q}{W}*100$ W is bacteria colonies of control and Q bacteria colonies of treated	Decreased cellular growth with the increasing content of nano-TiO_2_. With 0.8 g/L of nano-TiO_2_ there was a 98.5% of bacteria reduction	[96]
	*Staphylococcus aureus*	Gram- positive		Decreased cellular growth with the increasing content of nano-TiO_2_. With 0.8 g/L of nano-TiO_2_ there was a 99.0% of bacteria reduction	
Silver and Gold Nanoparticles on Ramie Fibers	*Escherichia coli*	Gram- negative	AATCC 100-2004 test: after exposing the fibers to the bacteria for 18 h at 120 rpm and 37 °C, the fibers are set aside and the cells are diluted, counted, and compared to the initial count.	Silver nanoparticles in ramie fiber showed 100% antibacterial activity because there was no growth of bacterial colonies on the culture plate.	[91]

**Table 4 molecules-27-07999-t004:** Antibacterial studies of sisal fibers.

Sisal Modification	Bacteria	Type	Method	Result	Ref
**Sisal Extracts**					
Sisal extract	*Escherichia coli*	Gram- negative	Disc diffusion—Qualitative method, absence or presence of a zone of inhibition in different concentrations	Significant inhibition, with zones or rings of inhibition of 10.69–12.17 mm. In addition, if the bacterial population is lower, the growth-inhibitory properties of the sisal extract could be improved	[103]
	*Bacillus* *stearothermophilus*	Gram- positive	Delvo test SP-NT kit—Broad spectrum microbial inhibition and antibiotic residues test—Colorimetric method: Purple color means to contain an inhibitor and Yellow color does not contain an inhibitor of microbial growth	100% inhibition—All samples were purple	
Aqueous Sisal extract	*Staphylococcus aureus*	Gram- positive	Two methods: Agar well diffusion method in vitro, absence or presence of a zone of inhibition in different concentrations. The test tube dilution method determines levels of resistance to an antimicrobial substance and measures minimum inhibitory concentration (MIC)	Significant inhibition, with zones or rings of inhibition of 29 mm. Its MIC of 10 mg/mL demonstrated significant biological activity against the test pathogenic organisms	[104]
	*Salmonella typhi*	Gram- negative		Significant inhibition, with zones or rings of inhibition of 27 mm. Its MIC of 20 mg/mL demonstrated a medium biological activity against the test pathogenic organisms	
**Non-Modified Sisal Fibers**					
Sisal fibers	*Escherichia coli*	Gram- negative	Microtiter plate biofilm assay semiquantitative assessment of the biofilm. Measure the optical density (OD) and classify it into strong, moderate, or weak biofilm creators	Microtiter plate biofilm assay semiquantitative assessment of the biofilm. Measure the optical density (OD) and classify it into strong, moderate, or weak biofilm creators	[101]
**Modified Sisal Fibers**					
Sisal-based activated carbon fibers	*Escherichia coli*	Gram- negative	The flasks with fibers and bacteria were shaken at 37°C for different periods. Surviving bacteria in the solution after contact with samples were counted by the spread plate culture method. The survival amount of bacteria (CFU/mL) was counted from the colony formed on the medium.	Completely kill *E. coli* at the concentration of 5 × 10^8^ CFU/mL in 8 h	[105]

**Table 5 molecules-27-07999-t005:** Antibacterial studies of cotton fibers.

Cotton Modification	Bacteria	Type	Method	Result	Ref
**Non-Modified Cotton Fibers**					
Naturally brown-colored cotton	*Escherichia coli*	Gram- negative	Antibacterial activity evaluated by AATCCTM100-2004 with the use of ethylene oxide in the sterilization process	Fibers showed antibacterial activity of 86.9%	[112]
	*Staphylococcus aureus*	Gram- positive		Fibers showed antibacterial activity of 91.7%	
**Modified Cotton Fibers**					
CuO-Cotton Fiber	*Escherichia coli*	Gram- negative	The viable bacteria were monitored by counting the number of colony-forming units from the appropriate dilution on nutrient agar plates	Treatment for 1 h with the coated cotton leads to the complete inhibition of *E. coli* and *S. aureus* growth.	[113]
	*Staphylococcus aureus*	Gram- positive			
Poly (sulfobetaine-acrylamide-allyl glycidyl ether) onto cotton	*Escherichia coli (ATCC8739)*	Gram- negative	Antibacterial properties were quantitatively evaluated by the viable cell count method	Functionalized cotton fibers exhibited a high level of antibacterial rate effect of 95.18%	[114]
	*Staphylococcus aureus (ATCC6538)*	Gram- positive		The modified fibers have efficient antibacterial properties of 98.74%	
TiO2 Nanoparticles-Coated Cotton Fabrics	*Escherichia coli*	Gram- negative	Antibacterial activity was qualitatively evaluated by the shake-flask method under visible light	Exhibited excellent antibacterial results because nearly no bacteria grew, then the antibacterial reduction rate reaches more than 99%	[115]
	*Staphylococcus aureus*	Gram- positive			

**Table 6 molecules-27-07999-t006:** Antifungal studies of linen fibers.

Linen Modification	Fungi	Method	Result	Ref
Modified Linen Fibers				
Linen fiber with didecyldimethylammonium nitrate—[DDA][NO_3_]	*Aspergillus niger van* Tieghem, *Chaetomium globosum* Kunze, *Gliocladium virens* Miller, *Paecilomyces variotii* Bainier, *Penicillium ochrochloron* Biourge	Evaluation of fungal growth measured according to the EN 14119, 2003 Standard	Showed antifungal activity. [DDA][NO_3_] applied in amount of 0.5% per 1 g of the dry fabric caused visible inhibition of mold growth on the fabric surface.	[118]
Pulp made from Linen fibers	*Aspergillus terreus Ate456*, *Aspergillus niger Ani245*, *Fusarium culmorum Fcu761*	Antifungal evaluation against the growth of fungi. The inhibition zones were reported.	Samples had positive effects against the growth of *A. terreus* and *A. niger*	[117]

## Data Availability

Not applicable.

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
