# Peer review of "Antimicrobial Properties of Plant Fibers"

_molecules, 2022, doi:10.3390/molecules27227999_

Round 1

Reviewer 1 Report

The author of this review introduced a comprehensive study about the utilization of plant fibers in biomedical applications like medical textiles, orthopedics, and dental implants. Although the review includes interesting materials, it can be accepted only after major revision.

Natural and synthetic fibers should be compared, and their advantages and disadvantages should be accounted. 

The authors in detail described various bacteria which can be distributed in biomedical devices but did not pay attention to viruses and molds. I understand the main goal of the authors is to show the successful application of the natural fibers against bacterial cells, but a short notification about the possibilities of applying the natural fibers against viruses and molds would be very useful. Furthermore, the description of bacteria infection is too long and consists of general information.

It will be valuable to mention cotton fibers (seed-hair fiber of several species of plants of the genus Gossypium) or linen fiber from plants of the genus Linum. Has their fiber had any antimicrobic properties? 

The images of the analyzed plants and fibers should be added.

Information on the physicomechanical properties of the fibers should be added (at least tensile strength and Young’s modulus).

The chemical structures of the antimicrobial agents from natural fiber should be added.

Examples of the successful applications of natural fiber should be illustrated, and figures from other papers can be used with permission.

Looking in the future is absent in the conclusion. It should be highlighted that directions for the development of natural fibers in antimicrobial applications are the most prospective.

Finally, I suggest citations of works where similar issues were discussed.

https://doi.org/10.1016/j.cej.2022.137048

https://doi.org/10.1134/S0003683822050076 

Author Response

1. Natural and synthetic fibers should be compared, and their advantages and disadvantages should be accounted.

As well-recommended by Reviewer 1, a new Section 3 was added.

2. The authors in detail described various bacteria which can be distributed in biomedical devices but did not pay attention to viruses and molds. I understand the main goal of the authors is to show the successful application of the natural fibers against bacterial cells, but a short notification about the possibilities of applying the natural fibers against viruses and molds would be very useful. Furthermore, the description of bacteria infection is too long and consists of general information.

As well-appointed by Reviewer 1, an extensive description of bacterial pathogens was realized in the manuscript since the majority of nosocomial infections are attributed to bacteria, in particular the bacteria species described in the manuscript (i.e., E. coli, P. aeruginosa, S. aureus, and S. epidermidis). Although the description of bacterial infection is long and general, we do believe that it is important to clarify the importance of these infections to the Readers in order to fully understand the review context, especially for the public non-familiar with microbial pathogens. However, as suggested by Reviewer 1, we also added brief information on viruses and fungi in the revised manuscript (please check the new sections 2.5 and 2.6 of the revised manuscript.

3. It will be valuable to mention cotton fibers (seed-hair fiber of several species of plants of the genus Gossypium) or linen fiber from plants of the genus Linum. Has their fiber had any antimicrobic properties?

It was added the subsection 4.4. related to cotton fibers and 4.5. about linen or flax fiber

4. The images of the analyzed plants and fibers should be added

Due to the large number of different plants and fibers analyzed, we could not include an image of all of them. However, the name of the species and the references are available in the manuscript for specific interests

5. Information on the physicomechanical properties of the fibers should be added (at least tensile strength and Young’s modulus).

Information was included in the description of each fiber in the section 4.

6. The chemical structures of the antimicrobial agents from natural fiber should be added

The complete information about the bioactive compounds of each fiber is not available in the papers; we include information related to functional groups and secondary metabolites that could be responsible for the antibacterial effect.

7. Examples of the successful applications of natural fiber should be illustrated, and figures from other papers can be used with permission

Now, the article includes 2 figures. The first one is related to antibacterial activity of natural fibers against bacterium types, and the Figure 2 based on biomedical applications.

8. Looking in the future is absent in the conclusion. It should be highlighted that directions for the development of natural fibers in antimicrobial applications are the most prospective.

This information was presented in the conclusion "The opportunity to use natural resources to reduce infection rates in surgeries, implants, materials, and medical supplies and their application opens an opportunity for short-term needs such as for cutaneous applications. Despite the publications on the antibacterial properties of natural fibers in biomedical applications, few works have reached in vivo experimentation, which is a necessary step for future applications and for an adequate evaluation of the safety of natural fibers.

9. Finally, I suggest citations of works where similar issues were discussed

Done

Reviewer 2 Report

This paper gives broad overview of different sources of plant fiber with antimicrobial activity. However, as this is review paper in journal Molecules, it is not enough to write methods and obtained results but to explain  mechanism of action in more details. And moreover, it would be good to stress some special metabolites (molecules) responsible for the antimicrobial activity. Otherwise, this paper is out of  scope.

Author Response

1.      And moreover, it would be good to stress some special metabolites (molecules) responsible for the antimicrobial activity. Otherwise, this paper is out of scope.

The complete information about the bioactive compounds of each fiber is not available in the papers; we include information related to functional groups and secondary metabolites that could be responsible for the antibacterial effect.

Round 2

Reviewer 1 Report

The authors reasonedly answered all of my issues and I can recommend the paper in its present form.

Reviewer 2 Report

Manuscript is now acceptable in the revised form.